# From Foxtail Millet Husk (Waste) to Bioactive Phenolic Extracts Using Deep Eutectic Solvent Extraction and Evaluation of Antioxidant, Acetylcholinesterase, and α-Glucosidase Inhibitory Activities

**DOI:** 10.3390/foods12061144

**Published:** 2023-03-08

**Authors:** Chunqing Wang, Zhenzhen Li, Jinle Xiang, Joel B. Johnson, Bailiang Zheng, Lei Luo, Trust Beta

**Affiliations:** 1Faculty of Food & Bioengineering, Henan University of Science & Technology, Luoyang 471000, China; 2Henan International Joint Laboratory of Food Green Processing and Safety Control, Henan University of Science & Technology, Luoyang 471000, China; 3School of Health, Medical & Applied Sciences, Central Queensland University, Bruce Hwy, North Rockhampton, QLD 4701, Australia; 4Department of Food & Human Nutritional Sciences, University of Manitoba, Winnipeg, MB R3T 2N2, Canada

**Keywords:** foxtail millet husk, deep eutectic solvents, α-glucosidase inhibitory activity, AChE inhibitory activity, antioxidant activity, waste management

## Abstract

Foxtail millet husk (FMH) is generally removed and discarded during the first step of millet processing. This study aimed to optimize a method using deep eutectic solvents (DESs) combined with ultrasonic-assisted extraction (UAE) to extract phenols from FMH and to identify the phenolic compositions and evaluate the biological activities. The optimized DES comprised L-lactic acid and glycol with a 1:2 molar ratio by taking the total flavonoid content (TFC) and total phenolic content (TPC) as targets. The extraction parameters were optimized to maximize TFC and TPC, using the following settings: liquid-to-solid ratio of 25 mL/g, DES with water content of 15%, extraction time of 41 min and temperature of 51 °C, and ultrasonic power at 304 W. The optimized UAE-DES, which produced significantly higher TPC, TFC, antioxidant activity, α-glucosidase, and acetylcholinesterase inhibitory activities compared to conventional solvent extraction. Through UPLC–MS, 12 phenolic compounds were identified, with 1-*O*-*p*-coumaroylglycerol, apigenin-*C*-pentosyl-*C*-hexoside, and 1-*O*-feruloyl-3-*O*-*p*-coumaroylglycerol being the main phenolic components. 1-*O*-feruloyl-3-*O*-*p*-coumaroylglycerol and 3,7-dimethylquercetin were identified first in foxtail millet. Our results indicated that FMH could be exploited by UAE-DES extraction as a useful source of naturally derived antioxidants, along with acetylcholinesterase and α-glucosidase inhibitory activities.

## 1. Introduction

Minimizing waste production in the agricultural and food sectors has been a significant topic of interest in recent years [1]. The current processing methods for plant foodstuffs, including grains, fruits, and vegetables, tend to generate large amounts of waste materials, including husks, seeds, peels, and other by-products [2,3]. However, these agricultural waste materials may contain high levels of valuable bioactive compounds, making them prospective feedstocks for the extraction and recovery of key compounds, including polyphenols, flavonoids, pectin, and dietary fiber [4].

Foxtail millet has been cultivated for over 8000 years, and China is considered to be the place of origin [5]. Foxtail millet is widely planted and is one of the major grain crops in Northern China [6]. Foxtail millet husk (FMH) is the outer layer of the seed and is generally removed and discarded to be a by-product during the first step of the millet processing. However, FMH is one of the prospective sources of polyphenols and other active ingredients, as previous research has demonstrated that the husk and aleurone layers of most cereal grains showed the highest concentration of phytochemicals [7]. Phenolics in foxtail millet present anti-proliferation effects on HT-29 human colon adenocarcinoma cells, breast cancer cells, and HepG2 liver cancer cells [8,9]. The extract from millet husk has also been used to synthesize silver nanoparticles, and the silver nanoparticles had antibacterial activity [10]. In our previous research, we reported that the husk contained a significantly higher content of phenolics compared with the dehulled millet used for cooking [11]. Therefore, it is necessary to develop optimized procedures to extract these valuable phenolics, allowing for value-added utilization of FMH.

In the two decades since it was first used by Abbott [12], the use of deep eutectic solvents (DESs) had gained more and more traction as a novel extraction procedure that was more green and sustainable. This technique has also been widely used for extracting polyphenols from plants or plant waste products, such as tea leaves, saffron processing wastes, *Moringa oleifera* leaves, and chestnut-shell waste [13,14]. The main advantages of DESs over conventional organic solvents include their low cost, widespread availability, biodegradability, efficiency, and environmental friendliness [15]. However, future studies need to be performed on the toxicity of the extract before it can be used for food application [16]. Ultrasonic-assisted extraction (UAE) is a very good auxiliary extraction method, and it has the advantages of a shorter extraction time, improved extraction efficiency, and better environmental protection. Furthermore, the highly efficient combination of UAE and DES has also gained a lot of attention because of the efficiency in extraction of high yields of bioactive compounds from plant matrices [17].

However, investigations on the green extraction method and utilization of phenolics from FMH, including its phenolic components and their bioactivities, have not been reported. Consequently, the main objectives of this study are to (1) optimize a green and efficient UAE-DES method to extract the phenolics from FMH; and (2) determine the phenolic compositions, antioxidant activity, and acetylcholinesterase and α-glucosidase inhibitory activities of FHM phenolics extracted, using DESs.

## 2. Materials and Methods

### 2.1. Raw Material and Reagents

Foxtail millet husk (dry) was acquired from Jinsu Agricultural Technology Corporation (Henan Province, China). The sample was smashed (RS-FS1401 mill, Royalstar Co., Ltd., Hefei, China) sieved (40-mesh), and then stored at −20 °C.

Betaine and L-lactic acid were obtained from Shanghai Macklin Biochemical Co., Ltd., (Shanghai, China). Acetylcholinesterase, *α*-glucosidase (from *Saccharomyces cerevisiae*), and standards (vanillic acid, p-hydroxybenzoic acid, p-hydroxybenzaldehyde, caffeic acid, p-coumaric acid, ferulic acid, syringic acid, rutin, and kaempferol, the purity above 98%) were sourced from Shanghai Yuanye Biotechnology Co., Ltd., (Shanghai, China), while ABTS(2,2-azinobis-(3-ethylbenzothiazoline-6-sulfonate)), DPPH(2,2′-diphenyl-1-picrylhydrazyl), Folin–Ciocâlteu reagent, Trolox (6-hydroxy-2,5,7,8-tetramethylchromane-2-carboxylic acid), and TPTZ(2,4,6-tris(2-pyridyl)-s-triazine) were acquired from Sigma-Aldrich Chemical Co. (St. Louis, MO, USA). Chromatographic methanol and formic acid were purchased from Thermo Fisher Scientific Reagent Co., Ltd., (Waltham, MA, USA). All other reagents, including glycerol, sodium carbonate (Na_2_CO_3_), sodium acetate, sodium hydroxide (NaOH), sodium nitrite (NaNO_2_), glycol, aluminum trichloride (AlCl_3_), ferric chloride (FeCl_3_), and methanol, were purchased from Tianjin Deen Chemical Reagent Co., Ltd. (Tianjing, China).

### 2.2. Preparation of DES

The preparation of DESs followed the method reported by Zheng et al. [18]. A range of different hydrogen bond donors (HBDs) and hydrogen bond acceptors (HBAs) were investigated in different molar ratios, as detailed in Table 1. To prepare each DES, the combined HBA and HBD was stirred continuously at 80 °C, until the solution developed a transparent, uniform appearance.

### 2.3. Extractions of Phenolic Compounds

A total of 0.2 g of milled FMH was combined with 2 mL of the DES (with 20% water content) in a 10 mL centrifuge tube. Extraction was conducted by sonicating the centrifuge tubes for 30 min (50 °C temperature; 300 W ultrasonic power). The supernatants, which included the extracted polyphenolic compounds, were obtained by centrifuging the extracts at 8000 rpm for 10 min.

For comparison, the ultrasonic-assisted conventional solvent extraction used the same process of UAE-DES but with 80% methanol solution as the solvent.

### 2.4. Experimental Design of Optimizing the Extraction Process

#### 2.4.1. Single-Factor Experiments

The single factors were water content of DES, extraction time, liquid-to-solid ratio, ultrasonic power, and extraction temperature, and all were explored by single-factor experiments. The ranges and gradients of each factor were selected from preliminary experimental work. The water content of the DESs ranged from 10 to 35%, the liquid-to-solid ratio ranged between 10 and 35 mL/g, the water-bath temperature varied from 30 to 80 °C, the extraction time was trialed between 10 and 60 min, and the ultrasonic power was set at 200 to 450 W.

#### 2.4.2. Response Surface Methodology (RSM)

To explore the interactions of these independent variables on TPC and TFC, an RSM experiment was conducted. This used a 4-factor, 3-level Box–Behnken design. The four independent variables were the DES water content (X_1_), extraction temperature (X_2_), extraction time (X_3_), and ultrasonic power (X_4_). Responses Y_1_ and Y_2_ were the TPC and TFC of the extracts, respectively.

Regression equations were calculated using the software of Design Expert 8.0.6 (11.0.1.0 64-bit, State-East Corporation). Regression coefficients were calculated from the experimental results, using the second-order polynomial model (Equation (1)). The performance of the constructed models was evaluated through a range of parameters, including their F-value, *p*-value, R^2^ value (coefficient of determination), and R^2^adj and R^2^pred values (adjusted and predicted coefficient of determination, respectively) [19]. A desirability function approach was used to calculate the optimal input variable values in order to maximize the yield of TPC and TFC. Finally, the theoretical results were verified by conducting triplicate extractions, using the optimal conditions, and comparing the experimental and predicted TPC and TFC.
(1)$$\gamma =\beta_{0}+\overset{k}{\underset{j=1}{\sum }}\beta_{j}x_{j}+\overset{k}{\underset{j=1}{\sum }}\beta_{jj}x_{j}^{2}+\underset{i}{\sum }\overset{k}{\underset{<j=2}{\sum }}\beta_{ii}x_{i}x_{j}+e_{i}$$

### 2.5. Determination of TPC and TFC

The TPC of the extracts was quantified by the Folin–Ciocâlteu colorimetric method, according the method of Zhang et al. [11]. The results were expressed in milligrams of ferulic acid equivalents (FAE) per gram of the FMH sample (mg FAE/g).

The TFC was measured by the aluminum chloride colorimetric method, as outlined by Xiang et al. [20]. The TFC was calculated as mg of rutin equivalents (RE) per gram of sample (mg RE/g).

### 2.6. Measurement of Antioxidant Activity

#### 2.6.1. DPPH Radical Scavenging Activity

The measure of the radical scavenging DPPH activity of the FMH extracts was conducted using our existing methods [21]. The results were expressed in micromoles of Trolox equivalents (TE) per gram of sample (μmol TE/g).

#### 2.6.2. ABTS^+^ Scavenging Activity

About ABTS^+^ free-radical scavenging activity, the extract was measured by a method previously described [21]. Again, results were expressed as μmol TE/g of sample.

#### 2.6.3. Ferric Reducing Antioxidant Power (FRAP)

To gain a full picture of a sample’s antioxidant activity, radical scavenging activity alone may not be sufficient. Consequently, the extracts’ reducing capacity was also quantified by the FRAP assay described by Zhang et al. [11], with results quantified as μmol TE/g of sample.

### 2.7. Inhibition of Acetylcholinesterase (AChE) Activity

The ability of the extracts to inhibit AChE activity was evaluated by the reported method [18]. Briefly, various concentrations of polyphenolic extracts were combined with 200 mmol/L phosphate buffer at pH 7.7 (200 μL), DTNB (80 μL), and 2 U/mL AChE enzyme (10 μL). After 5 min (at 25 °C), 15 μL of acetylthiocholine iodide substrate was added, the samples were left to sit for another 5 min, and then the absorbance measured at 405 nm. AChE inhibitory activities were expressed as IC_50_ values, using the calculated inhibition percentage, ***I*** (%) (Equation (2)).
(2)$$I=\frac{\left(A_{background}-A_{blank\;control}\right)-\left(A_{sample}-A_{sample\;control}\right)}{A_{background}-A_{blank\;control}}$$
where *A_background_, A_blank control_*, *A_sample_*, and *A_sample control_* are the absorbances measured for 100% enzyme activity (enzyme and solvent), the blank group (enzyme only), the test sample group (enzyme and sample), and the sample control group (sample only), respectively.

### 2.8. Inhibition of α-Glucosidase Activity

The α-Glucosidase inhibitory activity was determined with minor modifications to the method described by Li et al. [22]. In brief, various concentrations of polyphenol fractions (20 μL) were combined in a 96-well microplate with α-glucosidase enzyme (5 U/mL; 20 μL) and sodium phosphate buffer (0.2 M at pH 6.9; 120 μL). After 15 min of incubation (37 °C), the pNPG substrate (2.5 mM; 20 μL) was added before further incubation (10 min). Finally, 0.2 M sodium carbonate solution (80 μL) was added to terminate the reaction, before the resultant absorbance was measured at 405 nm. The results were calculated and expressed as IC_50_ values.

### 2.9. Identification and Quantification of Phenols by UPLC–MS/MS

To identify and quantify the phenols of the FMH extracts, exploratory profiling was conducted using the Waters H-Class UPLC system in series with a QqQ-MS (Waters Xevo TQ-S/micro). For the sample preparation before analysis, D101 macroporous resin was applied to separate the extract, according to method optimized by Zheng et al. [23]. The method used an Accucore C_18_ column (2.6 μm, 100 mm × 3 mm; Thermo Fisher Scientific, Waltham, MA, USA), injection volume with 10 μL, column temperature at 25 °C, and flow rate of 0.4 mL/min. The mobile phase comprised phase A (water) and phase B (chromatographic methanol), respectively, containing 0.1% formic acid, and used a 25-min gradient elution by the method of Yuan et al. [24]. The MS/MS information was collected in negative mode across the mass range of 100–1000 in a resolution of 5000. Phenolic components were identified with authentic standards where available, or otherwise tentatively identified through comparison of the UV spectra and MS/MS information in the literature.

To quantify the polyphenolic compounds, their peak areas were used at specific wavelengths: 280 nm for *p*-hydroxybenzaldehyde, 320 nm for *p*-coumaric acid and ferulic acid, and 350 nm for flavonoids. If a standard was not available for a particular compound, it was quantified as the equivalent concentration of its corresponding aglycone or closest analogue. All phenolic concentrations were expressed as mg/g of sample.

### 2.10. Surface Morphology Analysis

The surface morphology of the FMH cell structure was analyzed by scanning electron microscope (SEM), as described in Zheng et al. [18]. Following the extractions using the optimized UAE-DES and the UAE conditions, the remaining residues were air-dried naturally before being mounted on aluminum stubs. After coating these with gold-palladium, the Hitachi TM3030Plus SEM (HITACHI High-Technologies Corporation, Tokyo, Japan) was used to visualize the surface morphology.

### 2.11. Statistical Analysis

All of the experiments were repeated three times. Consequently, the results are expressed as mean ± standard deviation. The data of results were analyzed in IBM SPSS Statistics (Version 25.0, IBM Corp., Armonk, NY, USA) by the analysis of variance (ANOVA), followed by Duncan’s multiple range test and *t*-tests to determine statistically different results. A significance level of *p* < 0.05 was taken as statistically significant.

## 3. Results

### 3.1. Selection of DES

It has previously been found that, for phenolic acids and flavonoid compounds, sugar-based DESs have a lower extraction efficiency compared to amide, acid, and alcohol-based DESs [12,15,25]. According to the pre-experiments, the same DES showed different effects on the extraction of phenolic acids and flavonoids, so the TFC and TPC of the extracts were used to evaluate the extraction efficiency.

The TPC and TFC obtained after extracting FMH polyphenolics using eight different types of DESs are shown in Figure 1. As anticipated, both the TPC and TFC of FMH extracts varied widely depending on the extraction solvents used. Consequently, it can be stated that the DESs of different chemical natures differed significantly in their phenolic acid and flavonoid extraction capacities [13]. It could be seen that the DESs 1, 2, 4, 5, 6, and 8 gave excellent results on TPC. For TFC, DES-4 and DES-8 showed better results. There was no significant difference between DES-4 and DES-8 on TPC and TFC. DESs based on organic acids have higher polarity than DESs based on polyols and sugars [26]. Solvent polarity can influence the extraction efficiency since more polar compounds, such as polyphenols, obtained better extraction performance with eutectic solvents containing organic acids, so DES-4 and DES-8 showed a good effect in TPC extraction. Compared with DES-8, which is made of sodium acetate and L-lactic acid, the viscosity of DES-4, comprising L-lactic acid and glycol with a 1:2 molar ratio, was lower, and the flowability was better. Therefore, DES-4 was designated as the optimal solvent for extracting FMH phenolics.

### 3.2. Results of Single-Factor Experiments

#### 3.2.1. Water Content of DES

One of the easiest to alter—but most influential—factors influencing the properties of a DES is its water content [27]. Figure 2A displays the change in the extraction efficiency of phenolics for DESs with a different water content. The results of TPC and TFC showed a similar trend, beginning with an increase in yield between 10 to 15% water content, followed by a moderate decrease after 15%. The significant initial increase (*p* < 0.05) of extraction yield of phenolics might be attributable to the weakening of the hydrogen bonds. The increased amount of water in the DES could decrease the solvent viscosity and increase extraction yield [28]. Conversely, the significant decreases (*p* < 0.05) in extraction yield between 15 to 35% water content were likely a consequence of reduced solubility of moderately polar phenolics in the DES matrix with increasing polarity [29].

#### 3.2.2. Liquid-to-Solid Ratio

In contrast to the results observed for water content, the influence of liquid-to-solid ratio on TPC and TFC showed an asymptotical relationship (Figure 2B). With the increase of liquid-to-solid ratio, the TPC and TFC increased significantly up to 25 mL/g. However, there was no significant increase between 25 and 35 mL/g (*p* > 0.05), potentially indicating that maximum extraction efficiency had been reached. A similar trend was also observed by previous researchers extracting flavones from soy [30]. With the increase of the liquid–solid ratio, the dissolution amount of solute in solution will increase. Therefore, the phenolic concentration of the DES extract increased with the increase of the liquid–solid ratio. When the liquid–solid ratio increased to a certain extent, the dissolution number of phenolic compounds reached the maximum. Using an excess of extraction solvents produces unnecessary solvent waste and may actually hinder the recovery of phenolics; hence, a 25 mL/g liquid-to-solid ratio was selected as optimum and used in the following experiments.

#### 3.2.3. Extraction Time

The third variable investigated was extraction time. Initially, the increase of the extraction time provided a continuous and almost-linear increase in TPC and TFC (Figure 2C). This indicates that the ultrasonic treatment provided effective disruption of the FMH cell walls and internal vacuoles, allowing the polyphenols to diffuse out of the cells and into the solvent solution. Furthermore, during these shorter extraction times, there would be minimal diffusion resistance acting to prevent the polyphenols from exiting the intracellular environment. However, as the extraction time was past 40 min, the TPC and TFC started to decrease, possibly due to decomposition of some of the unstable polyphenol components when subjected to longer extracting times and relatively higher temperature [18].

#### 3.2.4. Extraction Temperature

The extraction temperature also affected the TPC and TFC yield significantly, as presented in Figure 2D. As the temperature increased, the TPC and TFC yield initially rose before both reached a maximum at 50 °C and fell off after this peak. Another study using acidified water to extract anthocyanins from blue honeysuckle berries reported similar results, with the highest yield at a temperature of 40 °C [28]. Higher temperatures are likely to increase polyphenol solubility in the DES and also increase the diffusion process of polyphenols from the intracellular environment into the DES. However, after a certain point, then the heat-sensitive polyphenols will begin to decompose due to the high temperature.

#### 3.2.5. Ultrasonic Power

Figure 2E shows an increasing trend in TPC and TFC as the ultrasonic power was raised from 200 to 300 W, reaching the highest TPC and TFC at 300 W power. Ultrasonic-assisted extraction uses ultrasound waves to induce localized pressure and cavitation regions in the matrix, which helps break up cell structures and release the phenolic compounds into solution. Consequently, it would be anticipated that increased ultrasonic power would give rise to an increase in extraction efficiency. However, as with extraction temperature, increasing the ultrasonic power beyond 300 W led to a reduction in TPC and TFC, again most likely due to the degradation of the polyphenols through a combination of direct ultrasonic energy and the higher water bath temperatures associated with ultrasonic activity [28]. It is worth noting that the TFC did not show as sharp a decline as TPC with increasing ultrasonic power, potentially indicating the greater stability of this compound class.

### 3.3. Fitting the Model

Table 2 shows the results for the 29 runs of TPC (Y_1_) and TFC (Y_2_) under different extraction conditions. To check the fit of the regression equations and quadratic polynomial model of yield, *t*-tests and an ANOVA were used, respectively. The *p*-value represented the significance of the variable; the smaller the *p*-value expressed, the more significant the impact of the variable on the results. The F-values were used to assess the relative contribution of each factor to the TFC and TPC yield [31], as shown in Table 3. The 3D plots and corresponding contour plots created based on the model are given in Figure 3 for directly displaying the effects of significant interaction terms on the responses of TPC and TFC. In the 3D diagram, the inclination of the surface is related to the influence of two interactive factors on the response value. The higher the inclination, the more significant the interaction between the two. The values on each curve in the response surface contour plots are the same. The color of the graph changes from blue to red, indicating that the value changes from small to large.

#### 3.3.1. Total Polyphenol Content

There was a significant positive correlation between the TPC response values and those calculated by the regression model detailed in Equation (3) (*p* < 0.001). As can be seen in Table 3, three factors had significant linear impacts on TPC (*p* < 0.05), namely the extraction time (X_3_), extraction temperature (X_2_), and ultrasonic power (X_4_) (in order of significance). Furthermore, four investigated factors all showed significant quadratic effects on the resultant TPC (*p* < 0.001). The only significant (*p* < 0.05) interactive effects on TPC were found for the X_2_X_3_ and X_2_X_4_ interaction terms. The non-significant factors were removed, and the predicted values of TPC were calculated using Equation (3).
(3)$$\begin{array}{cc} Y_{1}=7.23+0.53 & X_{2}+0.70X_{3}-0.13X_{2}X_{3}-0.40X_{2}X_{4}-0.26X_{1}^{2}-0.24X_{2}^{2}\\ & -0.25X_{3}^{2}-0.24X_{4}^{2}\; \end{array}$$

The variance analysis of the response surface displayed very high correction coefficients of R^2^ = 0.970 and R^2^_Adj_ = 0.939; the R^2^ were in reasonable accord with the R^2^_Adj_ (the difference is less than 20%), without significant lack of fit in the Equation (3) (*p* > 0.05). These data indicated that the model results were accurate; thus, it could be applied to predict TPC results when using DES-based UAE on foxtail millet husk [19].

In the polynomial Equation (3), the interaction term of X_2_X_3_ had a negative impact on TPC values (*p* < 0.05), indicating that, for short extraction times, higher temperatures gave a higher TPC, whereas higher temperatures lowered the TPC at longer extraction times. The interaction of these two factors (X_2_X_3_) can be seen in the 3D and contour plots in Figure 3A and Figure 3B, respectively. Similarly, the TPC was significantly negatively related to the X_2_X_4_ interaction (*p* < 0.05), suggesting that, with increasing ultrasonic power, the TPC gradually increased (within a certain temperature range). However, exceeding this range, the TPC gradually decreased with increasing ultrasonic power. The 3D plot and corresponding contour plot showing the interaction of temperature and ultrasonic power (X_2_X_4_) are presented in Figure 3C and Figure 3D, respectively.

#### 3.3.2. Total Flavonoid Content

As detailed in Table 3, the results from the ANOVAs indicated significant linear (X_1_ and X_3_), quadratic (X_1_^2^, X_2_^2^, X_3_^2^, and X_4_^2^) and interactive (X_1_X_4_ and X_2_X_4_) effects on TFC. Based on the regression coefficient (F) values, quadratic terms (X_1_^2^, X_2_^2^, X_3_^2^, and X_4_^2^) revealed the major effects, which were followed by X_1_, X_3_, X_1_X_4_, X_2_X_4_, and X_4_. The non-significant items were removed, and the formula for predicting TFC values is given in Equation (4).
(4)$$\begin{array}{cc} Y_{2}=4.18+0.17 & X_{1}+0.50X_{3}-0.25X_{1}X_{4}-0.14X_{2}X_{4}-0.25X_{1}^{2}-0.26X_{2}^{2}\\ & -0.25X_{3}^{2}-0.27X_{4}^{2}\; \end{array}$$

As observed for TPC, the developed regression model (Equation (4)) was strongly correlated with the TFC values (*p* < 0.001). The response surface variance analysis displayed high correction coefficients of R^2^ = 0.925 and R^2^_Adj_ = 0.920, with a good equation fit (*p* > 0.05 for lack of fit). Subsequently, this supported the use of this polynomial model for analyzing and predicting TFC extraction efficiencies using UAE-DES.

The interaction of X_1_X_4_ and X_2_X_4_ showed significant negative effects (*p* < 0.05) on TFC, meaning that the TFC gradually increased with increasing water content across a certain range of water contents, but exceeding this range, the TFC gradually decreased with the increase of ultrasonic power. Similarly, the interaction of ultrasonic power and extraction temperature on TFC showed the same trend. Figure 3E,F present the 3D plot and matching contour plot for the interaction between the DES water content and the extraction ultrasonic power (X_1_X_4_) on TFC. In the same vein, the interaction of the extraction temperature and ultrasonic power (X_2_X_4_) is shown in the 3D and contour plots in Figure 3G,H, respectively.

#### 3.3.3. Experimental Validation of the Model

The function models enabled the simultaneous optimization for the four extraction variables to provide the highest TPC and TFC. Consequently, the highest theoretical extraction yield was predicted to occur by using the following settings: water content of 15%, ultrasonic power of 304 W, 51 °C extraction temperature, and 41 min extraction time. The desirability value was 0.826 within a specific range (0.6–1) acceptable. With the conditions, the predicted values were 7.25 mg FAE/g DW for TPC and 4.18 mg RE/g DW for TFC.

For the experimental model validation conducted using the optimized parameters for extraction outlined above, the TPC and TFC were determined to be 7.38 mg FAE/g DW and 4.30 mg RE/g DW. This demonstrated remarkable correlation between the experimentally derived and theoretically predicted values, confirming the predictive accuracy of the model. Furthermore, it supported the use of the optimized extraction protocol for extracting polyphenol compounds from FMH.

### 3.4. TPC, TFC, and Biological Activities In Vitro

Table 4 shows the comparisons between the antioxidant activity, TPC, TFC, α-glucosidase and acetylcholinesterase inhibitory activity of the extracts using the UAE-DES and UAE.

The UAE-DES method gave a significantly higher TPC and TFC compared to UAE, suggesting that the UAE-DES had a better extraction efficiency than UAE. It may be that the DESs exhibit a high degree of solubility for phenolic compounds due to their ability to form hydrogen bonds with these solutes, which can be dissolved to a greater extent under the assistance of ultrasound [29].

Table 4 also shows that the extracts obtained using the UAE-DES showed higher antioxidant activity than that of UAE across all three of the different in vitro antioxidant assays used in this work. The scavenging ABTS^·+^ capacity of the phenolic extract by UAE-DES was 13.99 μmoL TE/g, almost twice as much as that of the UAE method, and the results of DPPH and FRAP assays presented similar situations. Generally, there was a significant positive correlation between the extractable phenolic content and the antioxidant activity of the sample [32,33]. As a result, the antioxidant activity of the UAE-DES extracts, which presented a significantly higher TPC and TFC, was higher than that of the extracts obtained byUAE.

In recent years, AChE inhibitors have gained increasing attention, and it has been found that some natural polyphenol compounds have certain inhibitory effects on AChE [34,35]. As shown in Table 4, the IC_50_ of the UAE-DES and UAE extracts were 295.53 μg/mL and 403.51 μg/mL, respectively. The lower IC_50_ of the UAE-DES extracts suggested stronger inhibitory AChE activity, likely due to the corresponding higher TPC and TFC [36]. The findings indicate that the polyphenol extracts from FMH have the ability to inhibit AChE activity and therefore can be utilized as a potential resource for natural AChE inhibitors.

Some chemical drugs, such as acarbose and voglibose, are widely applied to manage type II diabetes, mainly used to inhibit the activity of α-glucosidase; however, some side effects have been found in their applications [37]. Some studies have confirmed that *α*-glucosidase inhibitors were extracted from cereal products, which have fewer side effects [38,39]. The *α*-glucosidase inhibitory effect of the extracts from FMH was investigated, and the results are provided in Table 4. The IC_50_ values of the polyphenol compounds’ extracts were determined to be 190.12 μg FAE/mL and 280.22 μg FAE/mL by UAE-DES and UAE, respectively. The lower IC_50_ of the UAE-DES extracts suggested stronger inhibitory activity against *α*-glucosidase. The inhibitory activity against starch digesting enzymes are attributed to phenolic acids and flavonoids [40,41], and therefore the higher TPC and TFC actively resulted in the higher *α*-glucosidase inhibitory activity [42]. This might indicate that the abounding polyphenol compounds of FMH could be explored as one of potential sources for natural alternative products to manage type II diabetes.

### 3.5. Identification and Quantification of Phenolic Components

The mass spectral (MS^2^) information and UV spectral characteristics of those compounds were compared with the authentic standards or related references, and twelve polyphenol compounds were identified in the FMH extracts. The UPLC profile of polyphenol compounds of FMH at wavelength of 280 nm is shown in Figure 4. The retention time (RT), UV spectral characterizations, MS/MS fragments data, and contents of individual polyphenol compounds are listed in Table 5. The peaks were numbered by their elution times.

Compounds **1**, **2**, **3**, **4**, **5**, and **10** in the extracts were confirmed as *p*-hydroxybenzoic acid, *p*-hydroxybenzaldehyde, vanillic acid, chlorogenic acid, and ferulic acid, respectively, by comparing RT, UV spectrum and MS/MS fragments data with those of authentic standards. Compound **6** presented [M-H]^−^ at *m*/*z* 237 and the major MS/MS fragments at *m*/*z* 163,145, and 119. The daughter ion presented at *m*/*z* 119 was a typical fragment of *p*-coumaric acid produced by the loss of CO_2_ (*m*/*z* 44), and the predominant fragment at *m*/*z* 145 corresponded to the loss of glycerol (*m*/*z* 92), so compound **6** was identified as 1-*O*-*p*-coumaroylglycerol by comparing its mass spectra with previously reported results [18,43,44]. Compound **7** exhibited a molecular ion at *m*/*z* 593 and displayed typical fragmentations of *C*-glycosides 503 ([M-H-90]^−^), 473 ([M-H-90-30]^−^) and 353 ([M-H-90-30-120]^−^), and it was identified as apigenin-*C*-dihexoside. Compound **9** presented deprotonated molecular ions [M-H]^−^ of *m*/*z* 563, which was 30 amu lower than the deprotonated molecular ion of compound **7**. It also had typical *C*-glycoside fragmentations of 473 ([M-H-90]^−^) and 443 ([M-H-90-30]^−^), and therefore compound **9** was assigned as apigenin-*C*-pentosyl-*C*-hexoside. Compound **7** and compound **9** had been previously identified in millets [45]. Compound **8** exhibited an *m*/*z* signal at 436 and MS^2^ fragments at *m*/*z* 316, 273, 193, 145, and 119. It was identified as hydroxycinnamic acid amide, named N′, N″-di-*p*-coumaroylspermidine, which had been reported in foxtail millet and peanut flowers [18,45,46]. Compound **11** exhibited a deprotonated molecular ion [M-H]^−^ at *m*/*z* 413, and the main MS/MS fragments exhibited at *m*/*z* 237, 267, 163, and 193, showing the MS/MS signals of *p*-coumaroylglycerol, feruloylglycerol, *p*-coumaric acid, and ferulic acid; therefore, compound **11** was identified as 1-*O*-feruloyl-3-*O*-*p*-coumaroylglycerol, which has been reported in *Ananas comosus* L. leaves and *Lilium* [47,48]. To our knowledge, this is the first report to find 1-*O*-feruloyl-3-*O*-*p*-coumaroylglycerol in foxtail millet or its byproducts, and its UV spectrum and MS^2^ fragments are shown in Appendix A. Compound **12** presented [M-H]^−^ at *m*/*z* 329; the main MS^2^ fragments at *m*/*z* 314, 299, 285, 271 and 227; and the UV spectrum and MS^2^ fragments are shown in Appendix A. Therefore, compound **12** was identified as 3,7-dimethylquercetin, which is being reported in FMH for the first time but has been reported in seaweed [49].

The contents of individual polyphenol compounds are presented in Table 5. 1-*O*-feruloyl-3-*O*-*p*-coumaroylglycerol was the predominant phenolic component, as it showed the highest levels in both UAE-DES and UAE extracts, with the values of 890.27 μg/g and 508.20 μg/g, respectively. The other main phenolic components included apigenin-*C*-pentosyl-*C*-hexoside, 1-*O*-*p*-coumaroylglycerol, 3,7-dimethylquercetin, and *p*-coumaric acid. However, the contents of the individual phenolics in the extracts by UAE-DES were all significantly higher than those of the extracts by UAE. The sum of the individual polyphenol compounds in the UAE-DES extracts was far higher than that of the UAE extracts; the results were consistent with the TPC and TFC.

### 3.6. Scanning Electron Microscope Analysis

The aim of using scanning electron microscope to observe the microstructures of the FMH cells before and after the extractions is to observe the differences between the UAE-DES and UAE, and the results are shown in Figure 5. Compared to the observation result of the raw FMH sample (Figure 5A), the cell architecture of the FMH treated with UAE-DES (Figure 5B) was significantly damaged, and the sample surface showed many pores and cracks. In the case of the extract by UAE, the pores and cracks were smaller than those observed with UAE-DES, and some of the structure remained relatively undisturbed (Figure 5C). This may be due to the combined action of the cavitation from ultrasound and infiltration into the cell structure by DESs [50]. This lends further support to UAE-DES as a green and efficient method to extract the bioactive phenolic components from FMH.

## 4. Conclusions

The DES composed of L-lactic acid and glycol in a 1:2 molar ratio was screened as a suitable solvent for extraction phenolic from FMH, and the optimized extraction conditions were as follows: liquid-to-solid ratio of 25 mL/g, DES with water content of 15%, extraction temperature of 51 °C, extraction time of 41 min, and ultrasonic power at 304 W. The extracts of FMH were composed of twelve phenolic compounds, and we found that p-coumaric acid, 1-*O*-*p*-coumaroylglycerol, apigenin-*C*-pentosyl-*C*-hexoside, 1-*O*-feruloyl-3-*O*-*p*-coumaroylglycerol, and 3,7-dimethylquercetin exhibited higher concentrations. The DES extracts had higher antioxidant, α-glucosidase, and acetylcholinesterase inhibitory activities than UAE ones. The SEM illustrated the mechanisms behind the efficient extraction of the optimized UAE-DES. Therefore, the UAE-DES is proposed as a green and efficient extraction method of phenolic compounds from FMH based on our findings. Moreover, the DES extracts from FMH could be explored as one of potential sources of biologically active and natural polyphenol.

## Figures and Tables

**Figure 1 foods-12-01144-f001:**
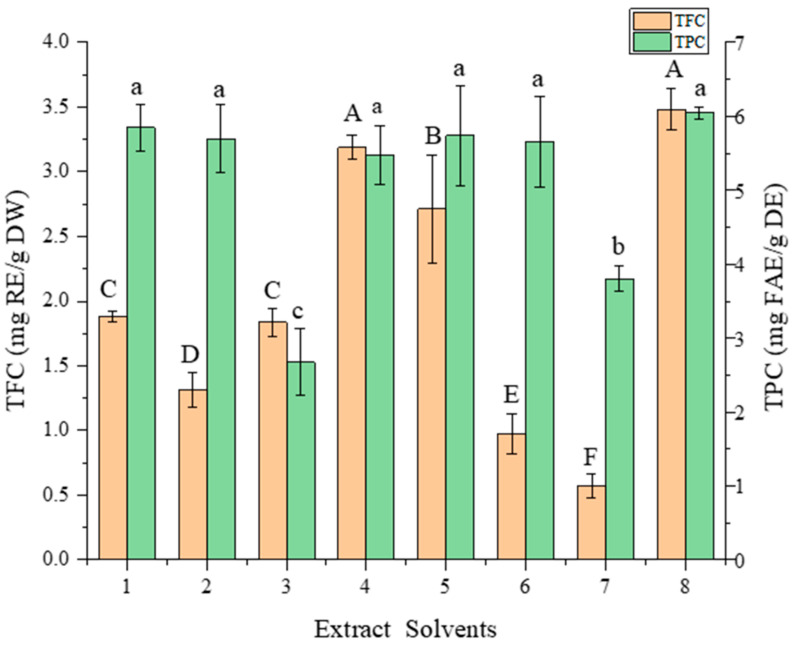
TFC and TPC of foxtail millet husk extracts extracted by different solvents. Different lowercase letters or uppercase letters indicate significant differences (*p* < 0.05).

**Figure 2 foods-12-01144-f002:**
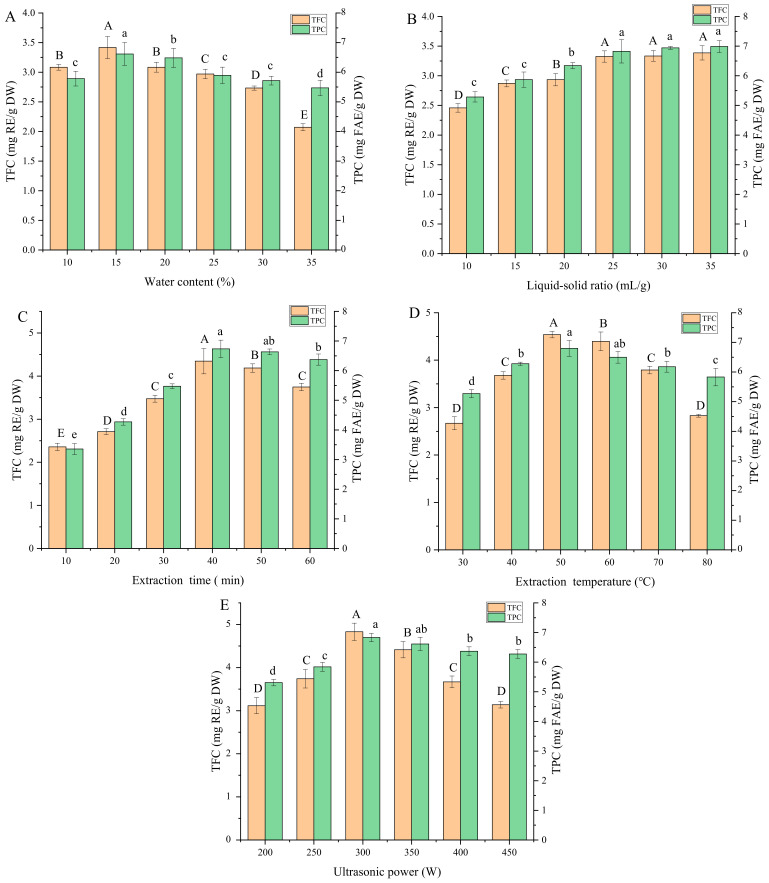
Effects of the investigated extraction variables on TFC and TPC from foxtail millet husk. water content (**A**), liquid–solid ratio (**B**), extraction time (**C**), extraction temperature (**D**), and ultrasonic power (**E**). Different lowercase letters or uppercase letters indicate significant differences (*p* < 0.05).

**Figure 3 foods-12-01144-f003:**
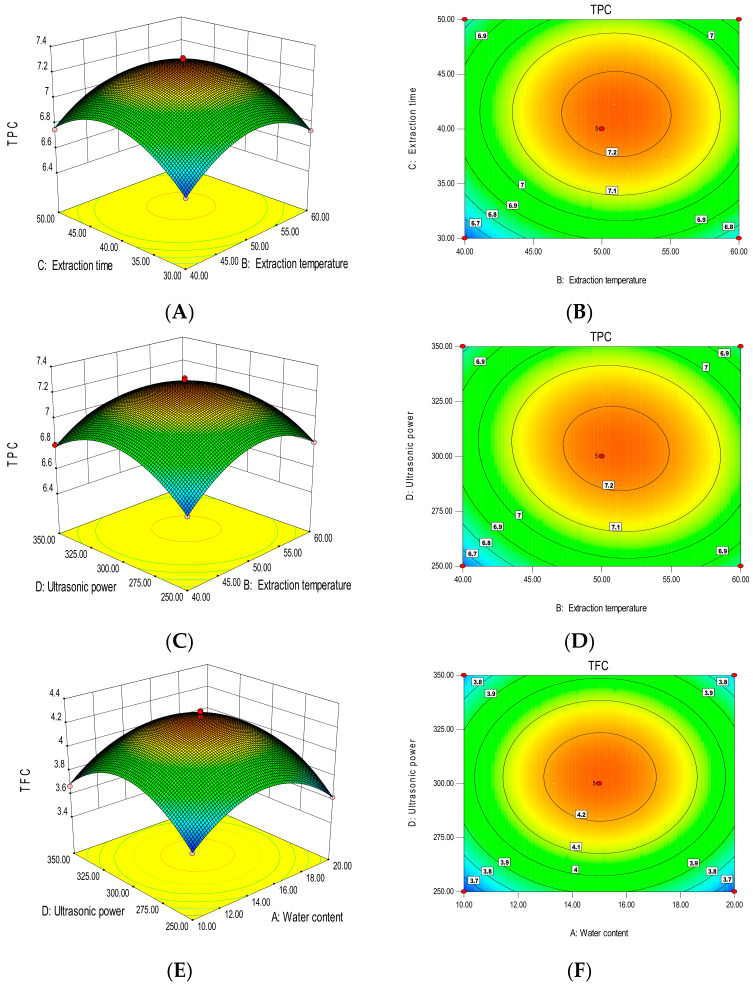
Three-dimensional response surface plots and corresponding contour plots. Influence of the extraction temperature and extraction time (**A**,**B**) and the extraction temperature and ultrasonic power (**C**,**D**) on TPC; the water content and ultrasonic power (**E**,**F**) and the extraction temperature and ultrasonic power (**G**,**H**) on TFC.

**Figure 4 foods-12-01144-f004:**
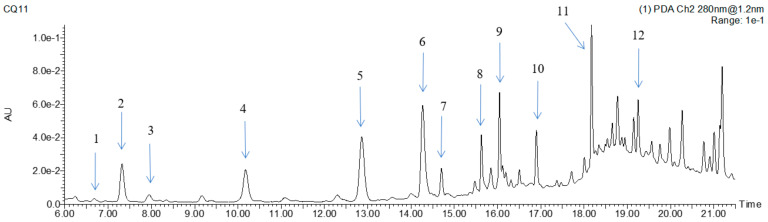
UPLC profile of polyphenol compounds of foxtail millet husk at 280 nm. The peaks of compounds were numbered by their elution times.

**Figure 5 foods-12-01144-f005:**
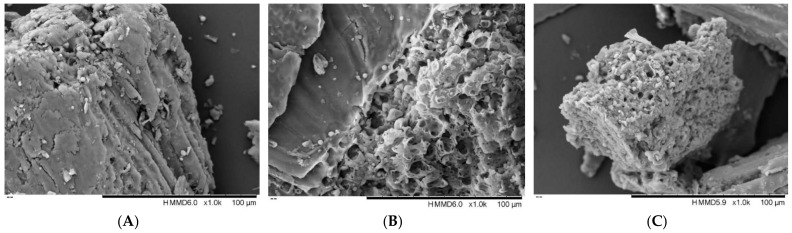
Scanning electron micrographs of foxtail millet husk before (**A**) and after extraction with UAE-DES (**B**) and UAE (**C**).

**Table 1 foods-12-01144-t001:** List of DES used to extract phenolics of foxtail millet husk.

No.	HBAs	HBDs	Molar Ratio
1	Betaine	Glycol	1:4
2	Betaine	Glycerol	1:4
3	Betaine	L-lactic acid	1:2
4	L-lactic acid	Glycol	1:2
5	L-lactic acid	Glycerol	1:2
6	Sodium acetate	Glycol	1:2
7	Sodium acetate	Glycerol	1:2
8	Sodium acetate	L-lactic acid	1:2

**Table 2 foods-12-01144-t002:** Experimental design and results of response surface methodology.

Run	Variables				TPC (Y_1_)	TFC (Y_2_)
	X_1_ (%)	X_2_ (°C)	X_3_ (min)	X_4_ (W)	(mg FAE/g DW)	(mg RE/g DW)
1	10	40	40	300	6.65 ± 0.04	3.60 ± 0.03
2	20	40	40	300	6.70 ± 0.05	3.66 ± 0.02
3	10	60	40	300	6.76 ± 0.01	3.68 ± 0.02
4	20	60	40	300	6.80 ± 0.02	3.67 ± 0.01
5	15	50	30	250	6.66 ± 0.03	3.57 ± 0.05
6	15	50	50	250	6.78 ± 0.02	3.66 ± 0.01
7	15	50	30	350	6.72 ± 0.01	3.58 ± 0.04
8	15	50	50	350	6.80 ± 0.03	3.79 ± 0.02
9	10	50	40	250	6.63 ± 0.03	3.60 ± 0.01
10	20	50	40	250	6.70 ± 0.01	3.62 ± 0.03
11	10	50	40	350	6.73 ± 0.05	3.66 ± 0.03
12	20	50	40	350	6.80 ± 0.02	3.68 ± 0.01
13	15	40	30	300	6.59 ± 0.01	3.58 ± 0.02
14	15	60	30	300	6.73 ± 0.06	3.61 ± 0.01
15	15	40	50	300	6.75 ± 0.04	3.66 ± 0.01
16	15	60	50	300	6.83 ± 0.02	3.73 ± 0.02
17	10	50	30	300	6.71 ± 0.03	3.63 ± 0.01
18	20	50	30	300	6.53 ± 0.02	3.69 ± 0.03
19	10	50	50	300	6.76 ± 0.01	3.72 ± 0.01
20	20	50	50	300	6.86 ± 0.04	3.78 ± 0.06
21	15	40	40	250	6.61 ± 0.01	3.57 ± 0.02
22	15	60	40	250	6.80 ± 0.03	3.67 ± 0.01
23	15	40	40	350	6.79 ± 0.10	3.70 ± 0.03
24	15	60	40	350	6.81 ± 0.01	3.74 ± 0.02
25	15	50	40	300	7.30 ± 0.01	4.11 ± 0.01
26	15	50	40	300	7.29 ± 0.02	4.23 ± 0.03
27	15	50	40	300	7.21 ± 0.05	3.92 ± 0.02
28	15	50	40	300	7.21 ± 0.01	4.32 ± 0.02
29	15	50	40	300	7.11 ± 0.01	4.31 ± 0.02

**Table 3 foods-12-01144-t003:** Analysis of variance for regression model equation.

Source	Response Variables			
	TPC		TFC	
	F-Value	*p*-Value	F-Value	*p*-Value
Model	31.75	<0.0001 ***	9.50	<0.0001 ***
X_1_	0.85	0.3730	14.37	0.0085 **
X_2_	12.69	0.0031 **	5.00	0.3332
X_3_	22.44	0.0003 ***	9.46	0.0440 *
X_4_	6.66 *	0.0218 *	5.38	0.2598
X_1_X_2_	7.39	0.9311 *	3.13	0.7266
X_1_X_3_	7.748 × 10^−34^	0.0166	2.836 × 10^−5^	0.9958
X_1_X_4_	3.826 × 10^−4^	0.9847	7.089 *	0.0491 *
X_2_X_3_	15.54	0.0077 **	5.63	0.8060
X_2_X_4_	12.04	0.0069 **	8.9 *	0.0369 *
X_3_X_4_	4.11	0.7015	2.63	0.8060
X_1_^2^	152.84	<0.0001 ***	44.42	<0.0001 ***
X_2_^2^	111.14	<0.0001 ***	50.64	<0.0001 ***
X_3_^2^	122.38	<0.0001 ***	67.73	<0.0001 ***
X_4_^2^	124.89	<0.0001 ***	55.17	<0.0001 ***
Residual	0.044		0.12	
Lack of Fit	0.021	0.22	0.015	0.056
Pure Error	0.024		0.11	
Cor Total	1.22		1.30	
R^2^	0.9695		0.9248	
Adj R^2^	0.9389		0.9196	
Pred R^2^	0.9074		0.8025	

* Significant at *p* < 0.05; ** significant at *p* < 0.01; *** significant at *p* < 0.001.

**Table 4 foods-12-01144-t004:** TPC, TFC, antioxidant activity, α-glucosidase, and AChE inhibitory activities of phenolic extracts from foxtail millet husk obtained using UAE-DES and UAE ^a^.

Parameter	UAE-DES	UAE
TPC (mg FAE/g DW ^b^)	7.38 ± 0.18 ^a^	4.40 ± 0.15 ^b^
TFC (mg RE/g DW)	4.30 ± 0.17 ^a^	2.63 ± 0.05 ^b^
ABTS^·+^ (μmoL TE/g DW)	13.99 ± 0.47 ^a^	7.59 ± 0.28 ^b^
DPPH (μmoL TE/g DW)	12.18 ± 0.14 ^a^	8.37 ± 0.33 ^b^
FRAP (μmoL TE/g DW)	18.36 ± 0.65 ^a^	9.88 ± 0.18 ^b^
IC_50_ of AChE (μg FAE/mL)	295.53 ± 1.56 ^a^	403.51 ± 2.42 ^b^
IC_50_ of α-glucosidase (μg FAE/mL)	190.12 ± 1.37 ^a^	280.22 ± 1.49 ^b^

^a^ Results are expressed as mean ± SD. Values with no letters in common are significantly different (*p* < 0.05). ^b^ DW, dry weight of sample.

**Table 5 foods-12-01144-t005:** Identification and quantification ^a^ of phenolic compounds in foxtail millet husk.

No.	Retention Time	[M-H]^−^ (*m*/*z*)	UV λ Max (nm)	Formula	*m*/*z* of Main Fragments (Relative Intensity, %), MS/MS	Identified Phenolic Compounds	Contents (μg/g DW ^b^)	
							UAE-DES	UAE
1	6.69	137	280	C_7_H_5_O_3_		*p*-hydroxybenzoic acid ^c^	45.32 ± 0.12 ^a^	30.22 ± 0.08 ^b^
2	7.32	121	284	C_7_H_5_O_2_		*p*-hydroxybenzaldehyde ^c^	146.21 ± 0.73 ^a^	87.32 ± 0.06 ^b^
3	7.96	167	287	C_8_H_7_O_4_		vanillic acid ^c^	68.32 ± 0.04 ^a^	42.13 ± 0.12 ^b^
4	10.19	353	320	C_16_H_18_O_9_		chlorogenic acid ^c^	110.00 ± 1.13 ^a^	63.10 ± 0.17 ^b^
5	12.92	163	308	C_9_H_7_O_3_	119(100)	*p*-coumaric acid ^c^	240.00 ± 1.23 ^a^	132.45 ± 1.36 ^b^
6	14.27	237	310	C_12_H_14_O_5_	237(5), 163(20), 145(100), 119(50)	1-*O*-*p*-coumaroylglycerol	395.75 ± 2.12 ^a^	245.01 ± 1.37 ^b^
7	14.70	593	271/328	C_27_H_29_O_15_	593(100), 503(10), 473(10), 353(10)	apigenin-*C*-dihexoside	176.95 ± 1.5 ^a^	92.01 ± 0.56 ^b^
8	15.63	436	270/331	C_25_H_30_N_3_O_4_	436(20), 316(55), 274(20), 195(25), 193(50), 145(20), 119(20)	N′, N″-di-*p*-coumaroyl spermidine	219.25 ± 1.13 ^a^	107.3 ± 1.30 ^b^
9	16.05	563	270/330	C_26_H_27_O_14_	563(100), 473(10), 443(10)	apigenin-*C*-pentosyl-*C*-hexoside	566.75 ± 3.38 ^a^	265.44 ± 1.22 ^b^
10	16.93	193	323	C_10_H_9_O_4_	193(20), 178(50), 134(100)	ferulic acid ^c^	55.35 ± 0.48 ^a^	38.17 ± 0.14 ^b^
11	18.17	413	313	C_22_H_22_O_8_	413(50), 397(10), 292(5), 267(10), 237(15), 193(100), 163(100), 119(30)	1-*O*-feruloyl-3-*O*-*p*-coumaroylglycerol	890.27 ± 6.14 ^a^	508.20 ± 4.45 ^b^
12	19.28	329	340	C_17_H_14_O_7_	329(28), 314(100), 299(60), 285(5), 271(20), 227(5)	3,7-dimethylquercetin	371.75 ± 1.86 ^a^	191.31 ± 1.33 ^b^

^a^ Results are expressed as mean ± SD. Values with no letters in common are significantly different (*p* < 0.05). ^b^ DW, dry weight of sample. ^c^ Identification of the compound was confirmed by the authentic standard.

## Data Availability

All related data and methods are presented in this paper. Additional inquiries should be addressed to the corresponding author.

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
