# Peer review of "From Foxtail Millet Husk (Waste) to Bioactive Phenolic Extracts Using Deep Eutectic Solvent Extraction and Evaluation of Antioxidant, Acetylcholinesterase, and α-Glucosidase Inhibitory Activities"

_foods, 2023, doi:10.3390/foods12061144_

Round 1

Reviewer 1 Report

This manuscript was written about deep eutectic solvents extraction, antioxidant, acetylcholinesterase, and α-glucosidase inhibitory activities of polyphenols from foxtail millet husk. This work showed extensive research, supported by various analytical data. However, there are some points below should be asked to the authors.

1.       What is the positive control of acetylcholinesterase, and α-glucosidase inhibitory assays? Please explain in the methodology section.

2.       Please check the sentence in the line 38.

3.       Please add some data about biological activities from recent publications such as Foxtail millet husk as an innovative biomass in the preparation of silica-silver composite with antimicrobial and free radicle scavenging activities, Materials Today: Proceedings, Volume 66, Part 4, 2022,1830-1836 in the introduction or discussion section.

4.       Please check Fig.3 and label Fig.3A, 3B, 3C and 3D.

5.       From the results of the antioxidant, acetylcholinesterase, and α-glucosidase inhibitory assays, the author was able to evaluate the potency of the extract or not. You can discuss this issue in the discussion or conclusion section to make the manuscript more interesting.

Reviewer 2 Report

The research article concerns the content of polyphenolic compounds and their biological activity contained in foxtail millet husk.

Below are some comments that may improve the value of this article:

- - Please add in the introduction more scientific information about the previously tested health-promoting properties of this plant.

- The following statement is not clear to me: '...investigations on the exploration and utilization of phenolics from FMH, including its phenolic components and their bioactivities, have not been reported’. After typing the phrase ‘foxtail millet husk‘ into the search engine, it searches for at least one article examining the milling fractions of foxtail millet (also husk) and their biological activities. For example ,https://www.sciencedirect.com/science/article/abs/pii/S0733521021000588?via %3Dihub (Comparative evaluation on phenolic profiles, antioxidant properties and α-glucosidase inhibitors effects of different milling fractions of foxtail millet). Please comment on your innovative approach.

- What was the purity of the standards and solvents used in the experiment? (especially using during UPLC)

- ‘As a measure of the radical scavenging activity of the extracts, the DPPH assay was conducted using our existing methods [18]’ – please add some details regarding the performance of this study. In the methodology that you originally followed, the ‘Effect of thermal processing on antioxidant properties of purple wheat bran’ measurements were performed several times at intervals. Now you have taken the measurement only after 30 minutes. This approach does not allow to monitor changes of absorbance over the time and when plateau will be achieved. If you have not decided on this type of measurement, why?

-  Please also add a larger description of ABTS+ radical scavenging activity and FRAP. Further descriptions of the performance of the experiment (nhibition of acetylcholinesterase (AChE) activity, inhibition of α-glucosidase activity) contain much more detail.

- Inhibition of α-glucosidase activity: ‘In brief, various concentrations of polyphenol fractions (20 μL) were combined…'. What exactly were the concentrations?

- Pages 22, 23, 24, 25 (lower drawings) (Fig. 3) - the numerical values are unclear. Please correct it. Perhaps darkening the font will give a satisfactory effect.

- Fig. 4. - Two arrows move away from the number 11.

- Please pay attention to correct punctuation. For example - between the quoted items of literature we enter a comma, not a semicolon.

Reviewer 3 Report

In the manuscript "From agricultural waste to bioactive phenolic extracts: Deep eutectic solvents extraction, antioxidant, acetylcholinesterase and α-glucosidase inhibitory activities of polyphenols from foxtail millet husk" the authors described the potential of coupling deep eutectic solvents and ultrasound assisted extraction for the extraction of bioactives from agricultural waste. The topic is interesting, but the manuscript should be improved. My suggestion is a major revision according to the listed comments:

Abstract: Could you please specify how DES was optimized?

Introduction: Can you clearly state the research's novelty?

Why was ultrasound-assisted extraction used? Please explain in the introduction section.

Line 69. It should be Moringa oleifera

Line 86. Please specify the storage conditions.

Line 220. Did authors measured physical properties of the prepared DESs?

Line 231. Please explain why the mentioned DESs showed good results for TPC extraction?

Figure 1. Please check the measuring units on the y-axes

Figure 2. Please improve the figure caption. Extraction conditions should be specified for each set of experiments.

Table 2. All results should be presented with standard deviations.

Section 3.3. Discussion should be improved.

Line 399. How were optimal conditions estimated?

Reviewer 4 Report

Find attached the file to improve the manuscript.

Round 2

Reviewer 1 Report

Thank you for the response. I have only a question about the control of α-glucosidase inhibitory assays. Do you have any control? I understand that you don't have positive control. Please show in the methodology if you have negative control to validate your experiment.

Author Response

Thank you for the response. I have only a question about the control of α-glucosidase inhibitory assays. Do you have any control? I understand that you don't have positive control. Please show in the methodology if you have negative control to validate your experiment.

Response: Blank control was applied in the experiment. We mainly compared the differences between the extracts obtained by the two extraction methods. While, comparing with the UAE, the extract from the UAE-DES method present higher α-glucosidase inhibition activity.

Reviewer 3 Report

The authors put significant effort to answered most of my questions, but one issue still must be clarified. As described in materials and method section, after series of single factor experiments Box-Behnken design of experiments was used. “To explore the interactions of these independent variables on TPC and TFC, a RSM experiment was conducted. This used a 4-factor-3-level Box-Behnken design. The four independent variables were: the DES water content (X1), extraction temperature (X2), extraction time (X3) and ultrasonic power (X4).” In my opinion results from those experiments should be presented in Table 2 with standard deviations and not only model predicted data. Therefore, my suggestion is still major revision.

Author Response

The authors put significant effort to answered most of my questions, but one issue still must be clarified. As described in materials and method section, after series of single factor experiments Box-Behnken design of experiments was used. “To explore the interactions of these independent variables on TPC and TFC, a RSM experiment was conducted. This used a 4-factor-3-level Box-Behnken design. The four independent variables were: the DES water content (X1), extraction temperature (X2), extraction time (X3) and ultrasonic power (X4).” In my opinion results from those experiments should be presented in Table 2 with standard deviations and not only model predicted data. Therefore, my suggestion is still major revision.

Response: We have added the standard deviations of the data in Table 2 according to your suggestion.

Reviewer 4 Report

Thanks for improving the manuscript. This is absolutely a better manuscript than the previous one. However, some revisions need to be done before publication. Based on your response to my comments, I still need some clarifications. I would be happy if you revised it based on these comments, and I am willing to review it again.

Comment 1

Rewrite the title. It should be "From foxtail   
millet husk (waste) to bioactive phenolic extracts using deep eutectic solvent extraction and evaluation of antioxidant, acetylcholinesterase, an α-glucosidase inhibitory activities.Comment 2.
Regarding the safety of DES, I agree that individual components have no toxicity. Yet, when they are mixed, research has shown some toxicity, so I suggest that you indicate that future studies need to be done on the toxicity of your extract before it can be used for food application. Below are some references.

Hayyan, M., Hashim, M. A., Al-Saadi, M. A., Hayyan, A., AlNashef, I. M., & Mirghani, M. E. S. (2013b). Assessment of cytotoxicity and toxicity for phosphonium-based deep eutectic solvents. Chemosphere, 93(2), 455–459. https://doi.org/10.1016/j.chemosphere.2013.05.013

Hayyan, M., Hashim, M. A., Hayyan, A., Al-Saadi, M. A., AlNashef, I. M., Mirghani, M. E. S., & Saheed, O. K. (2013c). Are deep eutectic solvents benign or toxic? Chemosphere, 90(7), 2193–2195. https://doi.org/10.1016/j.chemosphere.2012.11.004

Boateng, I. D. (2022). Evaluating the status quo of deep eutectic solvents in food chemistry. Potentials and limitations. Food Chemistry. https://doi.org/10.1016/j.foodchem.2022.135079

Comment 3
a lot of grammatical errors. Please read through it thoroughly to correct it.
Comment 4
regarding the question about ultrasonic pulsation, I was asking you about the pulse on time and pulse of time. Please provide that and also the power density.
Comment 5
regarding the question, "So what are the sample preparation procedures before UPLC-MS/MS?" your response is incomplete. So how much of the extract was picked, and what solvent was added to it before subjecting it to UPLC-MS analysis? Was that sample centrifuged and filtered using what size? Please add all of them to the manuscript.
Comment 6.
Regarding the answer to the question, " Better convert this to the p-value for easy comprehension," your response is not convincing. The p-value is most important in the response surface. A p< 0.01 is far more significant than a p< 0.05, which helps us know which variables are incredibly significant. I suggest you add the p-value in addition to the F-value. Below are my references.

Akpabli-Tsigbe, N. D. K., Ma, Y., Ekumah, J. N., Osabutey, J., Hu, J., Xu, M., ... & Quaisie, J. (2021). Two-step optimization of solid-state fermentation conditions of heilong48 soybean variety for maximum chlorogenic acid extraction yield with improved antioxidant activity. Industrial Crops and Products, 168, 113565

Akpabli-Tsigbe, N. D. K., Ma, Y., Ekumah, J. N., Osabutey, J., Hu, J., Xu, M., & Johnson, N. A. N. (2021). Novel solid-state fermentation extraction of 5-O-caffeoylquinic acid from heilong48 soybean using Lactobacillus helviticus: Parametric screening and optimization. LWT, 149, 111809.

Comment 7
regarding the answer to the question "Based on what model? Desirability function? What were your DF settings (maximize the TPC and TFC extraction)? So what was the DF value you got, and what shows that the value was in the acceptable range." the authors did not answer my question. You should calculate the desirability value and know if it is within a specific range (0.6-1) acceptable. That is very important. Although you did verification, the desirability value is the most important. You can use your "design expert" or "Minitab" to calculate this value using maximizing the TPC and TFC extraction.

Boateng, I. D. (2023). Application of Graphical Optimization, Desirability, and Multiple Response Functions in the Extraction of Food Bioactive Compounds. Food Engineering Reviews, 1-20.

Comment 8

Better delete CSE and replace it with UAE since you stated it is UAE.
